# Oracle-RLAIF: An Improved Fine-Tuning Framework for Multi-modal Video Models using Reinforcement Learning from Ranked Feedback

**Derek Shi**                                                     *derekshi@stanford.edu*
*Stanford University, CA, USA*
*Lawrence Livermore National Laboratory, CA, USA*

**Ruben Glatt**                                                        *glatt1@llnl.gov*
**Christine Klymko**                                                  *klymko1@llnl.gov*
**Shubham Mohole**                                                    *mohole1@llnl.gov*
**Hongjun Choi**                                                       *choi22@llnl.gov*
**Shashank Kushwaha**                                               *kushwaha1@llnl.gov*
**Sam Sakla**                                                          *sakla1@llnl.gov*
*Lawrence Livermore National Laboratory, CA, USA*

**Felipe Leno da Silva**                                     *lenodasilva@microsoft.com*
*Lawrence Livermore National Laboratory, CA, USA*
*Microsoft, WA, USA*

**Reviewed on OpenReview:** *https://openreview.net/forum?id=RIRgnRicTa*

## Abstract

Recent advances in large video-language models (VLMs) rely on extensive fine-tuning techniques that strengthen alignment between textual and visual comprehension. Many implementations typically begin with supervised fine-tuning (SFT) followed by reinforcement learning from preference data to enhance video comprehension. However, as VLMs scale in parameter size, so does the cost of gathering enough human feedback. To make fine-tuning more cost-effective, recent frameworks have explored reinforcement learning with AI feedback (RLAIF), which replace human preference with AI as a judge. Current RLAIF frameworks rely on a specialized reward model trained with video narratives to create calibrated scalar rewards– an expensive and restrictive pipeline. We propose Oracle-RLAIF, a novel framework that replaces the trained reward model with a more general Oracle ranker which acts as a drop-in model ranking candidate model responses rather than scoring them. Alongside Oracle-RLAIF, we introduce $GRPO_{rank}$, a modified Group Relative Policy Optimization (GRPO) loss function that directly optimizes feedback with rank-aware advantages. Empirically, we demonstrate that Oracle-RLAIF consistently outperforms leading VLM fine-tuning methods when evaluated across various video comprehension benchmarks. Oracle-RLAIF paves the path to creating flexible and data-efficient frameworks for aligning large multi-modal video models with reinforcement learning from rank rather than score based reward models.

## 1 Introduction

Current multi-modal foundation models are capable of providing fluent image captioning, accurate video question answering, and video reasoning capabilities (Ahn et al., 2024; Liu et al., 2023; Li et al., 2025a). Typically, the state-of-the-art (SOTA) models are additionally trained following a two-step process. First, supervised fine-tuning (SFT) through human-annotated videos allows video language models (VLMs) to produce syntactically correct and relevant answers (Luo et al., 2026). Next, following SFT, a reinforcement-learning-from-human-feedback (RLHF) (Ouyang et al., 2022) phase uses human preference over possible

model outputs to further video comprehension in the VLM. Although human labeled responses provide quality ground-truth data, this method introduces substantial bottlenecks of labeling inefficiency and cost. As a result, a few works in the literature started to rely on AI judge models to replace human feedback–resulting in reinforcement-learning-from-AI-feedback (RLAIF) (Ahn et al., 2024). RLAIF consists of using a model capable of outputting a reward signal for answers such that it can be used for fine-tuning. However, for many tasks, building a model capable of generating consistent and grounded rewards for any arbitrary combination of prompts and outputs has proven challenging (Shen et al., 2024; Chen et al., 2024).

We address these limitations by proposing a more flexible RLAIF framework called Oracle-RLAIF. Instead of relying on a fully functioning reward model capable of scoring any combination of prompts and responses, we only require an *Oracle model* capable of ranking responses in order of quality. This makes our framework broadly applicable to a variety of scenarios, including fine-tuning with feedback from a general-purpose closed-source model (e.g. through a model API) or distilling knowledge from a larger legacy model into a smaller, more efficient model. Moreover, we introduce a GRPO extension we call $GRPO_{rank}$, which directly processes ranks from Oracle to effectively fine-tune the initial multi-modal model. Across multiple video evaluation datasets, our Oracle-RLAIF model directly outperforms previous state-of-the-art fine-tuned VLMs. Our contribution is three-fold and can be summarized as follows:

1. We introduce Oracle-RLAIF, a novel rank-based RLAIF framework, utilizing a drop-in Oracle ranker, relaxing the need of a fully-functional reward model for RLAIF fine-tuning;

2. We propose $GRPO_{rank}$, a rank-aware modification of GRPO, in which we directly process rank signals in the loss function;

3. We extensively validate our proposed rank-based framework by training and evaluating our Oracle-RLAIF VLM. When directly compared to current fine-tuning techniques, Oracle-RLAIF improves video comprehension performance across benchmark datasets.

## 2 Background

We begin by providing all necessary background information for multi-modal learning as well as fine-tuning techniques for video language models. Additionally, we include in-depth description of the framework we build off and the specific limitations we address.

### 2.1 Multi-Modal Learning

Multi-modal learning trains models that can comprehend inputs from various modalities such as vision, audio and language (Radford et al., 2021; Alayrac et al., 2022; Li et al., 2023a). In video language models (VLMs), this means generating textual responses conditioned on visual information. These models typically comprise a vision encoder, which converts raw images into feature representations aligned with text (Radford et al., 2021; Li et al., 2023a), and a large language model (e.g., LLaMA, GPT (OpenAI, 2023; Touvron et al., 2023)) that generates responses based on those features.

To enhance video understanding, models are further trained via supervised fine-tuning (SFT) on video–question–answer triplets, grounding generation in visual reasoning tasks such as action detection or temporal comprehension. In recent work, to further align models, reinforcement learning after SFT has become necessary to achieve SOTA video question answering performance (Liu et al., 2023; Ahn et al., 2024).

### 2.2 Reinforcement Learning techniques for Fine-Tuning Video Language Models

After extensive training with supervised learning on labeled datasets, the model is capable of outputting contextually relevant answers. However, extensive spatial and temporal understanding is still limited. Two main frameworks have previously been implemented in reinforcement learning. Reinforcement Learning from Human Feedback (RLHF) is the process of using human feedback to choose a preferred response from two distinct model outputs for a given query (Figure 1-top). This preference dataset is then used to train the

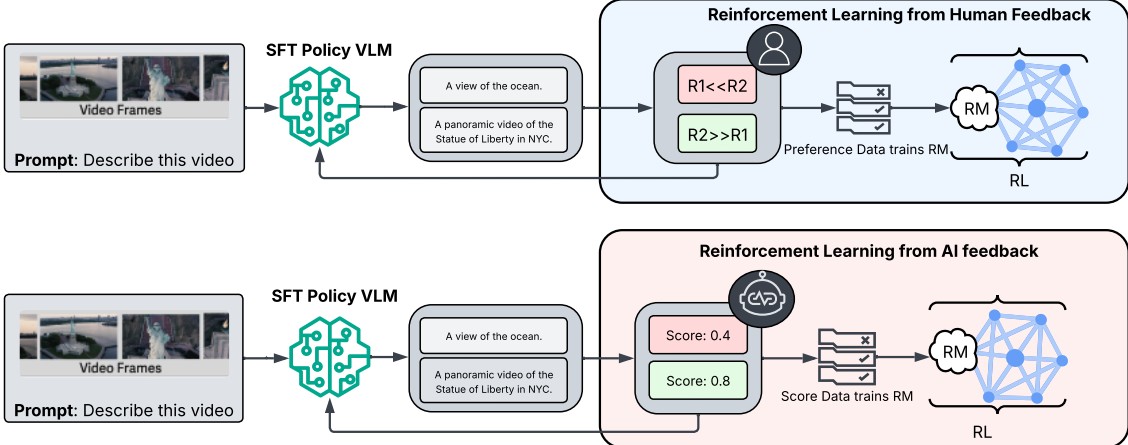

Figure 1: Dataset creation pipelines for RLHF (top) and RLAIF (bottom), illustrating how preferences are used to train reward models which are then used in the reinforcement learning loop to align the initial VLM.

reward model (RM) which is finally used in reinforcement learning to fine-tune the initial policy VLM (Ouyang et al., 2022). Due to the high cost of human labeling, researchers have begun using RLAIF (Bai et al., 2022; Lee et al., 2024; Su et al., 2023) where a *context aware AI* replaces the human as the judge (Figure 1-bottom). Recently, VLM-RLAIF applied this to multi-modal models by scoring two candidate responses from an initial supervised fine-tuned VLM (VLM-SFT) using a context aware AI judge. VLM-RLAIF significantly improved video question-answering abilities when compared to VLM-SFT, and currently achieves state-of-the-art performance across various video datasets (Ahn et al., 2024).

For both human and AI feedback, the purpose of the preference data is to train a reward model to score answers; and then use those rewards to guide an initial VLM towards preferred answers[1] (Hong et al., 2025). When in possession of this reward model, any canonical RL algorithm can be used to fine-tune the model responses. In the LLM fine-tuning space, a few specific algorithms are widely used since they have been demonstrated to be effective.

*Proximal Policy Optimization* (PPO) (Schulman et al., 2017) is arguably the most prominent algorithm for LLM fine-tuning, and is also the algorithm employed by the original VLM-RLAIF framework. PPO maximizes expected reward using a clipped surrogate objective that discourages large deviations from the previous policy, approximating a trust-region constraint while promoting stable policy gradient updates. The loss function for PPO is:

$$\mathcal{L}_{\text{PPO}}(\theta) = \mathbb{E}_t \Big[ \min\big(r_t(\theta)\,\hat{A}_t,\, \text{clip}\big(r_t(\theta), 1 - \varepsilon, 1 + \varepsilon\big)\hat{A}_t\big) \ - \beta\, D_{\text{KL}}\big[\,\pi_{\theta_{\text{old}}} \,\|\, \pi_\theta\big] \Big] \tag{1}$$

For a trajectory $\big(s_t, a_t\big)$ with return $\hat{R}_t$ and baseline value $V_\theta(s_t)$,

$$\hat{A}_t \ = \ \hat{R}_t - V_\phi(s_t), \qquad r_t(\theta) \ = \ \frac{\pi_\theta(a_t \mid s_t)}{\pi_{\theta_{\text{old}}}(a_t \mid s_t)} \tag{2}$$

The expected reward is represented in the advantage function $\hat{A}_t$, encoding how much better than the "average" action that particular action is expected to be. This policy optimization algorithm is used in VLM-RLAIF (Ahn et al., 2024) to guide the initial SFT model based on rewards from the trained RM.

---

[1]Some RLHF techniques do not require an explicit reward model in memory, however they follow assumptions regarding the preference distribution, thus the same general problem stands (Rafailov et al., 2023)

*Group Relative Policy Optimization* (GRPO) (Shao et al., 2024) addresses some limitations of PPO including unstable updates from reward magnitudes and the requirement of a trained value function to compute advantage. As introduced by DeepSeekMath, GRPO extends PPO to token probabilities $\boldsymbol{o_t}$ over $\boldsymbol{G_i}$ candidate responses for one query $\boldsymbol{q}$.

$$\mathcal{L}_{\text{GRPO}}(\theta) = \frac{1}{G} \sum_{i=1}^{G} \left[ \frac{1}{|o_i|} \sum_{t=1}^{|o_i|} \left\{ \min\left(r_t(\theta)\hat{\boldsymbol{A}}_{\boldsymbol{i,t}}, \ \text{clip}\left(r_t(\theta), 1-\epsilon, \ 1+\epsilon\right)\hat{\boldsymbol{A}}_{\boldsymbol{i,t}}\right) \right\} \right.$$

$$\left. - \beta \cdot D_{\text{KL}}\left[\pi_{\theta_{\text{old}}}(\cdot \mid \boldsymbol{q}) \,\|\, \pi_{\theta}(\cdot \mid \boldsymbol{q})\right] + c_{\text{entropy}} \cdot \mathcal{H}[\pi_{\theta}(\cdot \mid \boldsymbol{q})] \right] \tag{3}$$

where the policy importance ratio is defined as:

$$r_t(\theta) = \frac{\pi_\theta(o_{i,t} \mid \boldsymbol{q}, o_{i,<t})}{\pi_{\theta_{\text{old}}}(o_{i,t} \mid \boldsymbol{q}, o_{i,<t})} \tag{4}$$

and the advantage uses **normalized reward**

$$\hat{\boldsymbol{A}}_{\boldsymbol{i,t}} = \frac{R(\boldsymbol{q}, o_i) - \mu_R(\boldsymbol{q})}{\sigma_R(\boldsymbol{q})} \tag{5}$$

Since rewards are rescaled per-query, GRPO is robust to varying reward magnitudes and eliminates the need for a separate value function (Eq 5). Used by DeepSeek to finetune their R1 reasoning models, GRPO has already been proven effective for LLM post-training alignment (DeepSeek-AI, 2025).

*Direct Preference Optimization* (DPO) (Rafailov et al., 2023) is another popular algorithm used in LLM fine-tuning built to eliminate the need for a trained reward model. DPO replaces the reward function with preference loss, creating a binary cross entropy objective optimizing directly on preference pairs $(y_w, y_l)$.

$$\mathcal{L}_{\text{DPO}}(\theta) = -\mathbb{E}_{(x,y_w,y_l)\sim\mathcal{D}}\left[\log \sigma\left(\beta \log \frac{\pi_\theta(y_w \mid x)}{\pi_{\text{ref}}(y_w \mid x)} - \beta \log \frac{\pi_\theta(y_l \mid x)}{\pi_{\text{ref}}(y_l \mid x)}\right)\right] \tag{6}$$

## 3 RLAIF through Oracle Preferences

We propose the Oracle-RLAIF fine-tuning framework that replaces the traditional trained Reward Model (RM) with a drop-in Oracle ranker. We assume the Oracle ranker- whether a larger closed source model or a legacy system- can accurately rank the response quality conditioned on the prompt and visual context. Crucially, unlike frameworks such as VLM-RLAIF (Ahn et al., 2024), Oracle-RLAIF eliminates the need for auxiliary reward or value functions. Instead, it operates purely on the interaction between the initial Supervised Fine-Tuned (SFT) policy and the external ranker. By modifying the standard RLAIF loop to leverage direct rank-based supervision, the iterative fine-tuning process proceeds as follows (Figure 1 and Algorithm 1):

**1. Response Generation (Exploration)** For a given visual and textual prompt, the current policy (the SFT model) generates $N$ candidate responses. This allows the model to explore its own generation space.

**2. Oracle Ranking (Ground Truth)** A stronger multi-modal Oracle model (e.g. GPT-4o) evaluates the $N$ responses. It ranks them based on quality and relevance to the visual input. These rankings are treated as the ground truth.

**3. Policy ranking (Prediction)** The policy model calculates the internal log-probabilities for each of its own generated responses. These probabilities are sorted to produce the predicted rankings, representing the model's perceived confidence in its outputs.

**4. Rank-based Optimization** Our proposed loss function (Eq 7) compares the predicted rankings against the ground truth rankings. This signal updates the policy to align its internal confidence (log-probs) with the Oracle's external quality assessment.

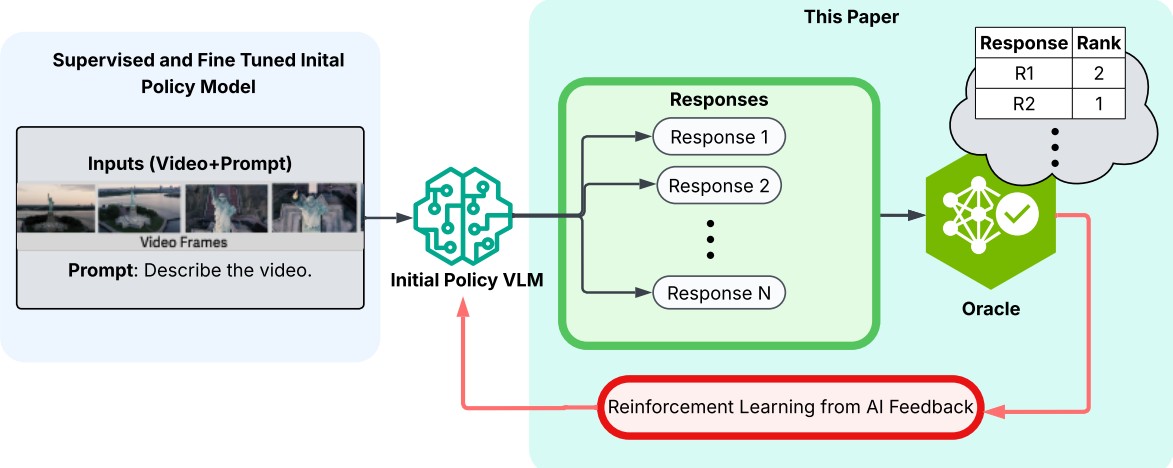

Figure 2: The general pipeline of fine tuning the initial policy VLM using Oracle-RLAIF. The light blue shading on the left indicates SFT training already applied to the initial policy VLM. The cyan shading on the right represents fine-tuning using our Oracle-RLAIF framework.

---

**Algorithm 1** Oracle-RLAIF

---

**Require:** Pretrained SFT Video Language Model $\pi_{\text{SFT}}$ (Policy Initialization)
**Require:** Oracle Model $\mathcal{O}_\phi$ (e.g., ChatGPT-4o)
**Require:** Multimodal Dataset $\mathcal{D} = \{(v, q)\}$ (Video, Question)
1: Initialize policy model: $\pi_\theta \leftarrow \pi_{\text{SFT}}$
2: **for** each prompt $(v, q) \in \mathcal{D}_b$ **do**
3:    **Step 1: Generation**
4:    Generate $G$ responses $\{o_i\}_{i=1}^{G}$ from $\pi_\theta(\cdot \mid v, q)$
5:    **Step 2: Ground Truth Ranking (Oracle)**
6:    Query $\mathcal{O}_\phi$ to rank $\{o_i\}$ based on quality and relevance to visual input $v$.
7:    Define ground truth ranking vector: $\mathbf{r}^* = [rank_{\text{oracle}}(o_1), \ldots, rank_{\text{oracle}}(o_G)]$
8:    **Step 3: Predicted Ranking (Policy)**
9:    Compute log-probabilities for each response under current policy:

$$\ell_i = \sum_{t=1}^{|o_i|} \log \pi_\theta(o_{i,t} \mid o_{i,<t}, v, q) \quad \forall i \in \{1, \ldots, G\}$$

10:    Derive predicted rankings $\hat{\mathbf{r}}$ by sorting $\boldsymbol{\ell}$ descending:

$$\hat{\mathbf{r}} = \text{argsort}_{\text{desc}}(\boldsymbol{\ell})$$

11:    **Step 4: Policy Optimization**
12:    Update policy parameters $\theta$ using a rank-alignment algorithm (e.g. $GRPO_{rank}$ 2) that minimizes the divergence between the predicted rank $\hat{\mathbf{r}}$ and oracle rank $\mathbf{r}^*$:

$$\theta \leftarrow \text{PolicyOptimizer}\left(\pi_\theta, \underbrace{\mathbf{r}^*}_{\text{Ground Truth}}, \underbrace{\hat{\mathbf{r}}}_{\text{Predicted}}\right)$$

13: **end for**
14: **return** final policy $\pi_\theta$

---

## 4 Fine-tuning Policy Models using Oracle Ranking

Training a model with PPO requires first training an intermediate Reward Model to map ordinal ranks into scalar pseudo-rewards, a process that can introduce significant instability due to, among other issues, insufficient generalization of the reward model. In contrast, we propose a modification of GRPO, $GRPO_{rank}$, to bypass this issue by directly turning the ranking input into a gradient, without the need of training an intermediate reward model. As our method draws from GRPO, we also inherit its advantages such as robustness in reward ranges.

### 4.1 Rank Adapted GRPO Objective

We introduce $GRPO_{rank}$ to optimize the policy within our Oracle-RLAIF framework. It applies non-linear penalization of rank errors, assigning larger penalties and smaller advantages proportionally to the deviation from the Oracle ground truth and the predicted rankings. It also penalizes false promotion of low-quality responses, encouraging the model to both prioritize high-quality outputs and suppress poor ones. To capture these intuitions, we use a normalized Discounted Cumulative Gain (nDCG) penalty that compares the *predicted rank* (the model's internal log probabilities) to the *ground-truth rank* assessed by the more powerful Oracle ranker. This penalty captures both position sensitivity and error severity, and is used to construct the advantage function for our policy gradient updates. For stability, $GRPO_{rank}$ generates on-policy responses while using a frozen reference policy for importance sampling. Our loss function modifies the original GRPO loss (Eq 3) by replacing the advantage term with $\hat{\boldsymbol{A}}_{\mathrm{rank}}$:

$$\mathcal{L}_{GRPO_{rank}}(\theta) = \frac{1}{G} \sum_{i=1}^{G} \left[ \frac{1}{|o_i|} \sum_{t=1}^{|o_i|} \left\{ \min \left( r_t(\theta) \hat{\boldsymbol{A}}_{\mathrm{rank}}, \ \mathrm{clip}\left( r_t(\theta), 1 - \epsilon, \ 1 + \epsilon \right) \hat{\boldsymbol{A}}_{\mathrm{rank}} \right) \right\} \right.$$
$$\left. - \beta \cdot D_{\mathrm{KL}} \left[ \pi_{\theta_{\mathrm{old}}}(\cdot \mid \boldsymbol{q}) \, \| \, \pi_\theta(\cdot \mid \boldsymbol{q}) \right] + c_{\mathrm{entropy}} \cdot \mathcal{H}[\pi_\theta(\cdot \mid \boldsymbol{q})] \right] \tag{7}$$

The advantage function $\hat{\boldsymbol{A}}_{\mathrm{rank}}$ in $GRPO_{rank}$ measures how much better each response $i$ is when compared to the model's "average response" and is defined as:

$$\hat{\boldsymbol{A}}_{\mathrm{rank}} = \mathbb{E}_{j \in G_i}[\delta_j] - \delta_i \tag{8}$$

with the expected group penalty for K responses within group $G_i$:

$$\mathbb{E}_{j \in G_i}[\delta_j] = \frac{1}{K} \sum_{j \in G_i} \delta_j \tag{9}$$

and the penalty term $\delta_i$ represents the deviation of the model's predicted rankings from the Oracle's ranking. This penalty is calculated using normalized Discounted Cumulative Gain (nDCG):

$$\delta_i = 1 - \mathrm{nDCG}_i = 1 - \frac{\mathrm{DCG}(\hat{\mathrm{rank}}_i)}{\mathrm{DCG}(\mathrm{rank}_i)} \tag{10}$$

where $\hat{\mathrm{rank}}_i \in \{0, \ldots, K-1\}$ is the model's predicted ranking computed from ordering the policy model's internal log probabilities and $\mathrm{rank}_i \in \{0, \ldots, K-1\}$ is from the Oracle ranker. Finally, DCG is computed as:

$$\mathrm{DCG}(\mathrm{rank}) = \frac{\frac{1}{1 + \mathrm{rank}}}{\log_2(2 + \mathrm{rank})} \tag{11}$$

---

**Algorithm 2** $GRPO_{rank}$ Optimization Step

---

**Require:** Policy $\pi_\theta$, Reference $\pi_{\text{ref}}$, Responses $\{o_i\}_{i=1}^G$
**Require:** Ground Truth Ranks $\mathbf{r}^* = [rank(o_1), \ldots, rank(o_G)]$ Predicted Ranks $\hat{\mathbf{r}} = [\hat{rank}(o_1), \ldots, \hat{rank}(o_G)]$

1: **1. Compute Rank Penalties (nDCG Deviation)**
2: **for** each response $i \in \{1, \ldots, G\}$ **do**
3: $\quad$ Calculate penalty $\delta_i$ based on deviation between $\hat{rank}_i$ and $rank_i$:

$$\delta_i = 1 - \frac{\text{DCG}(\hat{\mathbf{r}}_i)}{\text{DCG}(\mathbf{r}_i^*)} \quad \text{(Eq. 10)}$$

4: **end for**
5: **2. Compute Group Advantages**
6: Calculate average group penalty $\bar{\delta} = \frac{1}{G}\sum_{j=1}^G \delta_j$
7: **for** each response $i \in \{1, \ldots, G\}$ **do**
8: $\quad$ Compute advantage by subtracting individual penalty from group average:

$$\hat{A}_{\text{rank}}^{(i)} = \bar{\delta} - \delta_i \quad \text{(Eq. 8)}$$

9: **end for**
10: **3. Update Policy**
11: Update $\pi_\theta$ by minimizing the Rank-GRPO loss defined in Eq. 7

---

We detail the iterative $GRPO_{rank}$ fine-tuning algorithm in Algorithm 2. In each epoch, batches of prompts are sampled, and candidate responses are generated using the current policy model. The *Oracle model* provides relative rankings over responses in each group, from which nDCG-based penalties are computed. These penalties are then used to define an advantage function $\hat{A}_{\text{rank}}$, which captures how much better or worse each response is compared to the average performance in its group. The policy is updated by maximizing a clipped surrogate objective (Eq. 7), incorporating KL and entropy regularization to ensure stability. This allows the model to iteratively prefer responses ranked higher by the oracle while preventing large policy deviations.

### 4.2 Mathematical Properties

To further understand the motivation behind the elements in our $GRPO_{rank}$ objective, we highlight key mathematical properties that guided our formulation. This section explains the desired properties and how our formulation satisfies them. Additional examples from Appendix A.2 illustrate advantages via hypothetical rank configurations.

**1. Zero-Sum Property Within Groups.** Our advantage formulation:

$$\hat{A}_{\text{rank}} = \mathbb{E}_{j \in G_i}[\delta_j] - \delta_i$$

ensures that total advantage across group members sums to zero, enforcing relative comparison within groups:

$$\sum_{i \in G} \hat{A}_{i,t} = \sum_{i \in G} \left(\mathbb{E}_{j \in G}[\delta_j] - \delta_i\right) = K \cdot \mathbb{E}_{j \in G}[\delta_j] - \sum_{i \in G} \delta_i = K \cdot \left(\frac{1}{K}\sum_{j \in G}\delta_j\right) - \sum_{i \in G}\delta_i = 0$$

**2. Boundedness of Penalty.** Since $rank_i, \hat{rank}_i \in \{0, \ldots, K-1\}$, DCG values fall in:

$$\text{DCG}_i \in \left[\frac{1}{(1 + K - 1) \cdot \log_2(K + 1)}, 1\right]$$

Thus the normalized DCG, $\text{nDCG}_i = \frac{\text{DCG}_i(\hat{\text{rank}}_i)}{\text{DCG}_i(\text{rank}_i)} \in (0, 1]$, leading to bounded penalties:

$$\delta_i = 1 - \text{nDCG}_i \in [0, 1)$$

ensuring numerically stable and interpretable advantage values.

**3. Position-Sensitive Discounting.** The use of logarithmic discounting in:

$$\text{DCG}(\text{rank}) = \frac{\frac{1}{1+\text{rank}}}{\log_2(2 + \text{rank})} \tag{12}$$

ensures that rank errors at the top are penalized exponentially more than those at the bottom. This design choice explicitly encodes a prioritization to correctly rank the top answers, which are the responses actually shown to the user. Furthermore, our framework remains resilient to the inherent noise of lower-ranked samples. As noted by (Lambert et al., 2025), both reward models and LLM-as-a-Judge frameworks often collapse into high-variance "stochastic guessing" when forced to distinguish between nearly identical hard negatives. Thus, our penalties ($\delta$) are constructed as follows

$$\text{True Rank} = 0, \hat{Rank}_i = 1; \quad \text{nDCG} = 0.7925 \Rightarrow \delta = 0.2075$$
$$\text{True Rank} = 4, \hat{Rank}_i = 3; \quad \text{nDCG} = 0.9757 \Rightarrow \delta = 0.0243$$

Finally, the highest penalty originates from a low-ranked high-quality response; for example:

$$\text{True Rank} = 0, \hat{Rank}_i = 4; \quad \text{nDCG} = 0.5170 \Rightarrow \delta = 0.4830$$

## 5   Empirical Evaluation

The primary goal in our experimentation is to assess the effectiveness of Oracle-RLAIF compared to leading fine-tuning frameworks for VLMs– specifically VLM-RLAIF. We benchmark using datasets which test a model's capacity to interpret and describe visual events. To compare Oracle-RLAIF against VLM-RLAIF, we apply both frameworks to the same initial SFT policy model in (Figure 2), taken from the VLM-SFT 7B checkpoint as published in Ahn et al. (2024). Additionally, we compare against reported results of other strong baselines, assessing the contribution of our rank-based policy optimization approach.

**Evaluation Protocol**   Since our main purpose is to evaluate the fine-tuning performance of our proposal compared to VLM-RLAIF, we start both approaches with the same SFT model (the VLM-SFT 7B checkpoint as published in Ahn et al. (2024)), which is specifically of the `LLaMA-2 architecture`. From there, VLM-RLAIF is trained as described in their paper (Appendix A.1) with only a few training configurations changed as detailed below. We use their publicly released and pre-trained reward model to train VLM-RLAIF for 4 epochs and a rollout batch size of 64 for more frequent gradient updates (modified from 1 epoch and 256 batch size in the original publication to allow for more policy updates).

In reinforcement learning for Oracle-RLAIF, our initial SFT model outputs 5 candidate responses which are ranked by our Oracle ranker. In our implementation, the Oracle ranker is ChatGPT-4o. We acknowledge that using a closed-source Oracle limits reproducibility, so in Section 6.1 we run ablations with an open source Oracle model (PaliGemma2) and produced similar results. We provide the Oracle ranker with the same images our VLM uses to create responses, and the candidate responses related to the query. The Oracle model contains a comprehensive system message to strictly rank the candidate responses. These ranks are then fed into Rank-GRPO as the ground truth ranks which are compared to the internal model predicted ranks. Specifically, the sampling strategy we use is setting `Temperature=0.8` and `num_return_sequences = 5` in the respond function from the HuggingFace Transformers library. The number of training epochs is 2, over a dataset of 99k total videos with prompts.

A primary motivation of the Oracle-RLAIF framework is to enable fine-tuning of large video language models through reinforcement learning without the added complexity of a trained reward model. In this context,

Direct Preference Optimization (DPO) stands out as the prevalent standard for reward-model-free alignment, however we empirically show that DPO does not perform as well as Oracle-RLAIF. Therefore, we include a DPO baseline to directly compare to the effectiveness of Oracle-RLAIF using our proposed $GRPO_{rank}$ (Eq 7). To maintain consistency with Oracle-RLAIF, we generate *two outputs* from the initial model, and preference is provided by the same Oracle API ChatGPT-4o with identical system messages, and we call this baseline *Oracle-DPO*.

All models are trained using 4×NVIDIA H100 80GB GPUs with *Quantized Low-Rank Adapter (QLoRA)* (Dettmers et al., 2023), which enables efficient fine-tuning by combining 4-bit quantization with low-rank adapters. The fully trained models are then evaluated as described in the next subsections.

**Evaluation Datasets and Benchmarks**   We compared the models across two distinct evaluation regimes. In the first, we follow the evaluation pipeline used in VLM-RLAIF by benchmarking across MSVD, MSRVTT, and ActivityNet (Wu et al., 2017; Xu et al., 2016; Yu et al., 2019) video question answering datasets which target action recognition, temporal reasoning, and overall multi-modal understanding. Since these benchmarks use open-ended questions, it is standard practice to have the model's responses be evaluated by an LLM (in this case GPT-3.5-Turbo) in comparison to human-annotated ground-truth labels. This follows the methodology described in Li et al. (2023b). The LLM as a judge considers five key dimensions (response relevance to video content, detail capture, contextual understanding, temporal reasoning, and consistency) to create a numeric score from 0 to 5 as well as a binary "yes" or "no" indicating correct response. While using an LLM as a judge for evaluation can create variance, for these benchmarks specifically there is no better way to match model responses to the ground truth, as lexical matching would be inappropriate.

In the second evaluation regime, we instead benchmark against the better performing "final" checkpoint of the VLM-RLAIF 7B model as trained and evaluated by the authors[2]. In this comparison, we take advantage of a more contemporary dataset Video-MME (Fu et al., 2025) which was not available at the time of VLM-RLAIF's publication. By design, Video-MME does not originate from existing video evaluation datasets, allowing no possibility of data leakage into training. Additionally, unlike prior benchmarks which rely on LLM-judgment, Video-MME establishes objective and reproducible evaluation through multiple choice question answering. Video-MME is currently widely used for benchmarking, and we consider this our most meaningful experiment.

Table 1: Performance of Oracle-RLAIF versus baseline VLMs in zero-shot question answering across MSVD, MSRVTT, and ActivityNet.

| Model | Fine-tuning method | MSVD-QA | | MSRVTT-QA | | ActivityNet-QA | |
|---|---|---|---|---|---|---|---|
| | | Acc. | Score | Acc. | Score | Acc. | Score |
| VideoLLaMA ((Zhang et al., 2024)) | SFT | 51.6 | 2.5 | 29.6 | 1.8 | 12.4 | 1.1 |
| Video-ChatGPT ((Maaz et al., 2024)) | SFT | 64.9 | 3.3 | 49.3 | 2.9 | 35.2 | 2.7 |
| BT-Adapter ((Liu et al., 2024)) | SFT | 67.5 | 3.7 | 57.0 | 3.2 | 45.7 | 3.2 |
| Video-LLaVA ((Lin et al., 2024)) | SFT | 70.7 | 3.9 | 59.2 | 3.5 | 45.3 | 3.3 |
| LLaMA-VID ((Li et al., 2024)) | SFT | 69.7 | 3.7 | 57.7 | 3.2 | 47.4 | 3.23 |
| VideoChat2 ((Li et al., 2025a)) | SFT | 70.0 | 3.9 | 54.1 | 3.3 | 49.1 | 3.3 |
| **VLM-SFT (baseline)** | SFT | 67.2 | 3.6 | 52.4 | 3.0 | 44.1 | 3.2 |
| **VLM-RLAIF** | RLAIF (PPO) | 68.5 | 3.6 | 54.2 | 3.1 | 46.1 | 3.4 |
| **Oracle-DPO** | RLAIF (DPO) | 64.7 | 3.4 | 53.2 | 3.1 | 40.1 | 2.9 |
| **Oracle-RLAIF** | RLAIF ($GRPO_{rank}$) | **73.5** | **4.0** | **60.8** | **3.8** | **48.0** | **3.5** |
| $\Delta$ (Oracle − VLM-RLAIF) | | **+5.0%** | **+0.4** | **+6.6%** | **+0.7** | **+1.9%** | **+0.1** |

---

[2]Despite our best attempt to replicate the results in the original paper using the public implementation, a few key scripts were missing causing slightly lower performances than the final published checkpoint. In the second evaluation, however, we use the best performing model, even though we could not replicate its training.

**Detailed Analysis**  Table 1 shows the results for our first experiment[3]. Oracle-RLAIF outperforms all baselines we evaluated in video-question answering performance. Specifically, we observe consistent gains across all three benchmarks when compared to our reproduced VLM-RLAIF model following the author's framework. This demonstrates that our rank-based optimization approach yields more effective policy updates compared to VLM-RLAIF, which employs scalar reward modeling and PPO updates. The original VLM-RLAIF publication reports significantly better results than we could achieve on ActivityNet— which we attribute to the use of ActivityNet caption data in their reward model training which we omitted, thus creating unfair evaluation within this dataset. As for why Oracle-DPO underperforms SFT, we assert that DPO does not use the full range of candidate responses as $GRPO_{rank}$ does. Thus, decomposing it into a 1 to 1 comparison provides less information in each update. Given additional optimization steps, we expect DPO would eventually surpass SFT, but it lacks the sample efficiency of $GRPO_{rank}$ within the same training budget.

Table 2: Video-MME: performance comparison across short and medium videos

| Category | Oracle-DPO | VLM-RLAIF | Oracle-RLAIF | VLM SFT |
|---|---|---|---|---|
| Temporal (Perc. + Reas.) | 49.5% | 38.8% | **50.1%** | 48.2% |
| Action (Rec. + Reas.) | 34.8% | 29.6% | **38.1%** | 32.9% |
| Object (Rec. + Reas.) | 46.3% | 36.8% | **49.0%** | 39.5% |
| Spatial (Perc. + Reas.) | 41.4% | **45.1%** | 43.4% | 43.3% |
| Attribute Perception | 40.2% | 39.4% | **42.3%** | 39.5% |
| Counting Problem | 27.1% | 19.4% | **29.5%** | 19.6% |
| OCR Problems | **38.1%** | 34.6% | 37.8% | 34.5% |
| Information Synopsis | 48.5% | 49.4% | 48.0% | **50.1%** |
| Overall Accuracy | 41.4% | 36.9% | **43.2%** | 37.5% |

Therefore, in our second experiment reported in Table 2, we use Video-MME ensuring no data leakage from training to evaluation. Here, Oracle-RLAIF significantly outperforms the original VLM-RLAIF checkpoint, demonstrating superior generalization and robustness. We hypothesize that VLM-SFT slightly outperforms VLM-RLAIF because traditional reward models are prone to exploit low-level patterns such as response length or specific keywords rather than true video comprehension. Furthermore, as highlighted in (Lambert et al., 2025), many reward models struggle to distinguish between distinct "negatives" which appear superficially correct. Consequently, the RLAIF process with a trained RM may have inadvertently introduced noise or led to reward hacking, whereas the SFT baseline remains anchored to the human-curated distribution of the training data. Importantly, Oracle-RLAIF does not suffer from these Reward Model biases.

Overall, Oracle-RLAIF achieves a +6.3% percent improvement in average accuracy over VLM-RLAIF, with significant improvements in *Temporal Perception and Reasoning* (+11.3%), *Counting Problems* (+10.1%), and *Object Recognition and Reasoning* (+12.2%)– tasks which involve physical detection and causal events. For instance, Oracle-RLAIF accurately answers questions such as "*Which action is not included in the third magic trick?*" from *Action Recognition*, and successfully understands time-based cues in questions like "*In which part of the video is the woman in the blue top interviewed?*" from *Temporal Perception*. These results point to the core strength of our rank-based optimization to improve model alignment with temporally and causally grounded responses.

In contrast, Oracle-RLAIF shows performance declines in *Spatial Perception and Reasoning* and *Information Synopsis*. We hypothesize that these categories contain higher ambiguity or abstraction, causing rank-based optimization to be less effective. For example, *Spatial Reasoning* questions such as "*What holiday was the video most likely recorded during?*" or "*What can be inferred about the setting from the lighting and background?*" rely on implicit cues that are not effectively optimized through relative ranking. Furthermore, we hypothesize the slight divergence in improvement may stem from a scarcity of high-quality training samples in those specific domains. Similarly, tasks requiring spatial awareness across frames may benefit from further investigation into architectural adjustments. All together, further investigation is necessary to understand this

---

[3]We do not replicate experiments for all models that are not bolded, and instead report their published results from prior works.

drop in performance, however, the general performance gains in comparison to Oracle-DPO, VLM-RLAIF, and the VLM-SFT baseline strongly validate our framework's efficacy in multi-modal video understanding.

# 6 Ablation Studies

After running the previous experiments, we conducted further ablations to validate our claims. Specifically, we investigate replacing the Oracle ranker with both an open-source model and a trained reward model, and hyperparameters within $GRPO_{rank}$.

## 6.1 Replacing the Oracle Ranker with PaliGemma 2

To further substantiate our claim that the Oracle-RLAIF framework is widely flexible and cost efficient in comparison to training a reward model in VLM-RLAIF, we integrate open-weight model (PaliGemma 2 [10B]) as the Oracle ranker. Since PaliGemma is a much smaller model than GPT-4o, we simplified the original system prompt in order to produce successful rollouts. Similar to Table 2 we evaluate on short and medium length videos in the Video-MME dataset, and then take the average.

Table 3: Video-MME: Including PaliGemma 2 as Oracle

| Category | VLM-RLAIF | Oracle-RLAIF$_{GPT-4o}$ | VLM SFT | Oracle-RLAIF$_{GEMMA}$ |
|---|---|---|---|---|
| Temporal (Perc. + Reas.) | 38.8% | **50.1%** | 48.2% | 48.1% |
| Action (Rec. + Reas.) | 29.6% | **38.1%** | 32.9% | 33.8% |
| Object (Rec. + Reas.) | 36.8% | **49.0%** | 39.5% | 42.1% |
| Spatial (Perc. + Reas.) | **45.1%** | 43.4% | 43.3% | 39.2% |
| Attribute Perception | 39.4% | **42.3%** | 39.5% | 41.1% |
| Counting Problem | 19.4% | **29.5%** | 19.6% | 26.6% |
| OCR Problems | 34.6% | **37.8%** | 34.5% | 37.0% |
| Information Synopsis | 49.4% | 48.0% | **50.1%** | 45.8% |
| Overall Accuracy | 36.9% | **43.2%** | 37.5% | 39.9% |

As reported in Table 3, Oracle-RLAIF$_{GEMMA}$ achieves an overall higher performance than the SFT model but under performs using GPT-4o as the oracle. This demonstrates that performance gains can be achieved and reproduced even with smaller open source models, and as expected, performance gains scale alongside the capability of the Oracle model.

## 6.2 Replacing the Oracle ranker with Trained RM

By comparing against VLM-RLAIF, we already assessed the effectiveness of our Oracle ranker and $GRPO_{rank}$ relative to a learned reward model (RM) and PPO-based optimization. In this ablation, we replace the Oracle API with a trained reward model, in order to answer the important question: despite the significant computation overhead, does Oracle-RLAIF benefit from using a calibrated reward model, or is a legacy API sufficient as the ranking signal?

We adopt the reward model released from Ahn et al. (2024), which is trained to assign a scalar reward to candidate responses. To integrate this RM into our ranking-based RLAIF setup with $GRPO_{rank}$ (Eq. 7), we convert the scalar rewards into rankings by sorting candidates according to reward magnitude. During evaluation, we report accuracy across the same VQA benchmarks used in Table 1.

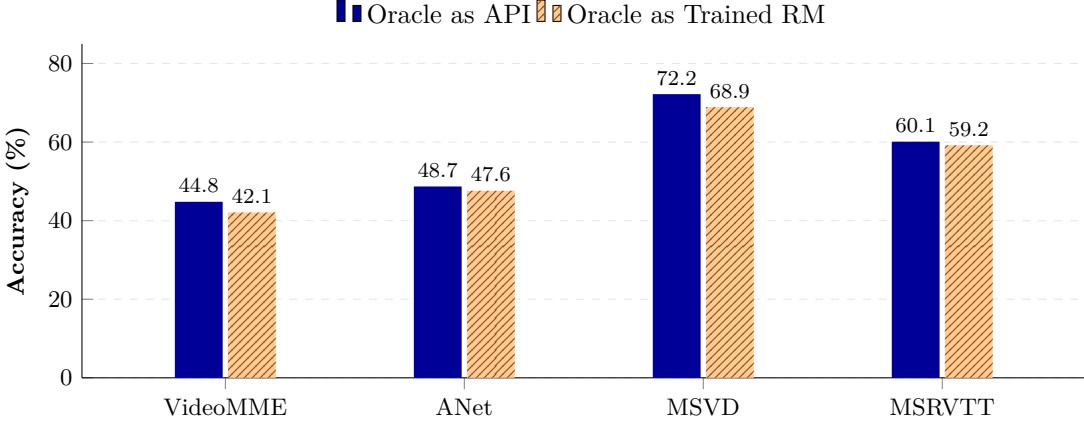

Figure 3: Comparison of Oracle-based alignment methods across video benchmarks.

As shown in Figure 3, the direct API-based Oracle consistently matches or exceeds the performance of the trained Reward Model across all benchmarks. This confirms our hypothesis that the intrinsic reasoning capabilities of a strong general-purpose API are sufficient to provide high-quality ranking signals. Therefore, $GRPO_{rank}$ achieves effective preference optimization without needing the additional computational overhead and complexity of training and calibrating a dedicated reward model.

### 6.3 Number of Return Sequences in $GRPO_{rank}$

In Oracle-RLAIF, a large computational overhead comes from the N candidate responses produced in the fine-tuning process. Thus, we hope to optimize the amount of responses to capture variances in output, while minimizing the cost as well. We vary the number of return sequences (RS) using 3, 4, 5, and 10 and report the corresponding performances on various VQA datasets.

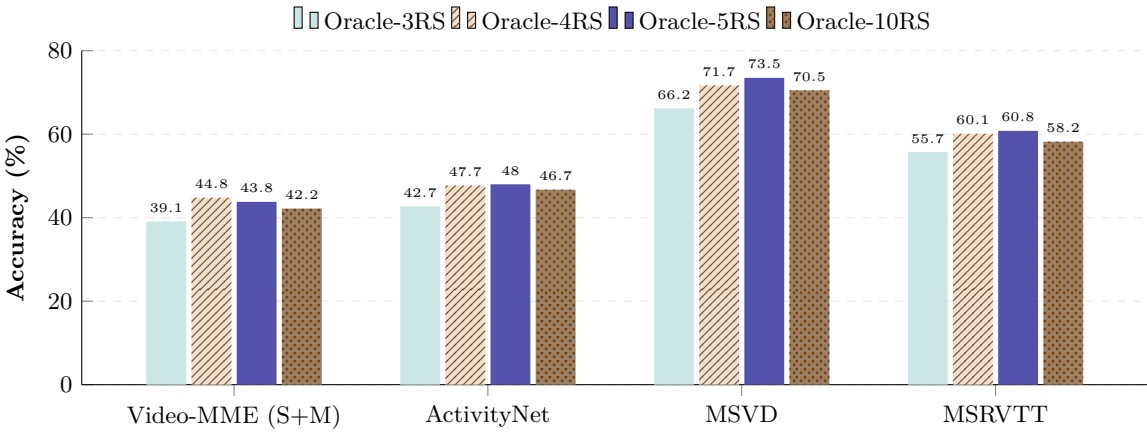

Figure 4: Performance comparison of Oracle-RS variants across benchmarks.

Figure 4 demonstrates that performance generally improves with an increased number return sequences from 3 to 5, but drops off when increased to 10. Therefore, we employ Oracle-5RS in our experiments to maximize absolute VQA accuracy; however, Oracle-4RS serves as a viable alternative where computational efficiency is prioritized, delivering comparable results with lower generation costs.

# 7 Related Works

Our work builds on a growing body of literature exploring alignment through preferences, rankings, and human feedback. The core innovation of Oracle-RLAIF lies in directly leveraging rank signal and hierarchical ordering in our proposed loss function, without any pre-processing or transformation. While prior work has utilized rating-based feedback (Luu et al., 2025; Xu & Zhu, 2025; Yuan et al., 2023; Choi et al., 2024), none formulate a loss function as we do. Existing approaches differ fundamentally in how they treat ranking information and structure their optimization objectives. For instance, ERL-VLM (Luu et al., 2025) uses rankings to *generate training data for a reward model* rather than to optimize the policy directly. Meta-RL (Xu & Zhu, 2025) maintains ranking across tasks within a meta-learning framework, instead of using group-wise ranks for per-update feedback. RRHF and RAFT (Yuan et al., 2023; Yao et al., 2024) apply ranking in supervised fine-tuning to align with human preferences but do not incorporate it into policy gradient methods. While RRPO (Sarkar & Etemad, 2025) attempts to stabilize training by considering relative reward differences, it still operates on a learned scalar reward boundary. Our work differs by moving entirely to a listwise ranking advantage. Furthermore, while TPO (Li et al., 2025b) explores trajectory-level preferences, it does not formulate a policy gradient update that directly maximizes listwise metrics like nDCG. Finally, recent specialized video models like LLaVA-Hound-DPO (Zhang et al., 2025) and VistaDPO (Huang et al., 2025) have successfully adapted offline preference learning to video-language models, mitigating temporal hallucinations, and similarly VideoPASTA (Kulkarni & Fazli, 2025) introduces spatio-temporal constraints for alignment. However, these methods primarily utilize pairwise DPO, which lacks the group-relative exploratory benefits of $GRPO_{rank}$. Importantly LLaVA-Hound-DPO, VideoPASTA, and TPO are fundamentally constrained by the need for pre-generated, domain specific preference pairs. The effectiveness of our framework comes from its ability to allow researchers to flexibly distill knowledge across diverse domains by directly projecting the Oracle's expertise.

# 8 Conclusion

In this work, we introduce a novel reinforcement learning from AI rankings framework to fine-tune large video multi-modal models. We use a drop-in Oracle ranking model in place of the trained and calibrated reward model used in past RLAIF frameworks. To handle ranked feedback, we develop a rank adapted Group Relative Policy Optimization algorithm $GRPO_{rank}$. We train and empirically evaluate Oracle-RLAIF across multiple video-text understanding benchmarks and directly outperform leading fine-tuning methods. While this study establishes the efficacy of our framework on the LLaMA-2 architecture, investigating scaling behavior on newer baseline models such as Qwen3-VL or InternVL3.5 is vital to confirming the universality of $GRPO_{rank}$. Furthermore, testing our framework directly against more contemporary preference-based optimization algorithms such as LiPO (Zheng et al., 2025) would help further our claim and effectiveness. Finally, we intend to explore the use of multi-modal oracles in the context of accelerating general RL tasks outside of video-language-answering (Silva et al., 2017; 2020).

## Acknowledgment

This work was performed under the auspices of the U.S. Department of Energy by Lawrence Livermore National Laboratory under Contract DE-AC52-07NA27344 and was supported by the LLNL-LDRD Program under Project No. 25-SI-001. LLNL-JRNL-2016123.

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

# A   Appendix

## A.1   VLM-RLAIF Specific Training Configurations

We build directly on the RLAIF framework by (Ahn et al., 2024) which has proven successful in fine tuning *video–language* models. Below we discuss their specific framework framework.

**Policy architecture**   Their policy starts from LLaMA-2-7B and inserts a frozen openai–clip-vit-large-patch14-336 followed by a 32-token Q-Former adapter, yielding $\approx 7.1$ B parameters. The vision encoder contains two linear layers and additional learnable parameters using LoRA (Hu et al., 2022) This creates a LLaVA framework as the pretrained model (Liu et al., 2023).

**Supervised warm-start**   Before RL, the model is SFT-pretrained on synthetically generated video-text instruction-tuning data (80k) (Maaz et al., 2024; Su et al., 2023), video question answering datasets (67k) (Xiao et al., 2021; Liu et al., 2023), and further generated object-centric datasets (180k). In SFT, the tuning dataset is split into easy (214k) and hard (113k) to create a curriculum learning strategy (Chang et al., 2021) which begins with basic video question answering before progressing to advanced comprehension question answering; where difficulty is measured as the correct answer length.

**Reward Model Trained with Context** In order to ground their reward model used to score responses for PPO, the authors trained using video, question, video narrative, and preference. This video narrative was gathered by prompting VLM-SFT over the ActivityNet dataset (more than 99,000 videos). The final 13B RM is trained to score responses, and used to drive reinforcement learning with PPO.

**Reinforcement Learning with PPO fine-tuning** In the VLM-RLAIF fine tuning pipeline, the initial policy outputs two responses per query and computes advantage $\hat{A}_t$ with a learned value head $V_\phi(s_t)$, and updates the policy using the clipped PPO objective and a frozen reference policy (Eq 1).

**Training Details** For input, videos are uniformly sampled to 50 frames, where the CLIP visual encoder extracts spatial and temporal features similar to (Maaz et al., 2024). In SFT, LoRA rank and $\alpha$ are set to 32 and training is for one epoch at each stage. For RL, QLoRA (Dettmers et al., 2023) rank is 65 and $\alpha$ to 16 and train the policy model for one epoch . All using 8xNVIDIA A100 GPUS (80G)

## A.2 Illustrative Examples

Below is a hypothetical table showing computed values for different predicted ranks when $K = 5$, ground-truth rank = 0:

Table 4: $\text{GRPO}_{\text{rank}}$ advantage calculation with ground truth rank = 0 and predicted ranks from 0 to 4

| Pred. Rank | DCG | nDCG | $\delta_i = 1 - \text{nDCG}_i$ | $\mathbb{E}[\delta_j]$ | $\hat{A}_{\text{rank}}$ |
|:---:|:---:|:---:|:---:|:---:|:---:|
| 0 | 1.0000 | 1.0000 | 0.0000 | 0.2887 | +0.2887 |
| 1 | 0.7925 | 0.7925 | 0.2075 | 0.2887 | +0.0812 |
| 2 | 0.6667 | 0.6667 | 0.3333 | 0.2887 | −0.0446 |
| 3 | 0.5805 | 0.5805 | 0.4195 | 0.2887 | −0.1308 |
| 4 | 0.5170 | 0.5170 | 0.4830 | 0.2887 | −0.1943 |

