# OpenReview forum: "Oracle-RLAIF: An Improved Fine-Tuning Framework for Multi-modal Video Models using Reinforcement Learning from Ranked Feedback"
_TMLR — Accepted by TMLR_

### Review · Reviewer_h6e6 · 2026-03-27

**Summary Of Contributions:**

The paper proposes Oracle-RLAIF, a reinforcement learning framework for fine-tuning video-language models (VLMs) using ranked feedback from an Oracle rather than scalar rewards from a trained reward model. The authors introduce GRPO_rank, a modification to GRPO that uses nDCG to penalize rank deviations and compute advantages. The framework is evaluated against PPO-based RLAIF and DPO baselines on select video benchmarks.

**Strengths**: Bypassing the need to train a dedicated multi-modal reward model simplifies the traditional RLHF/RLAIF pipeline. The integration of nDCG to formulate the advantage function is a reasonable approach to translating ordinal rankings into continuous signals required by policy gradient methods. The ablation comparing an API Oracle to a trained Reward Model Oracle usefully isolates the contribution.

**Audience:**

Yes

**Audience Explanation:**

The idea of replacing a trained reward model with a general-purpose Oracle ranker is a practically motivated direction, and the nDCG-based advantage formulation for listwise RL is of interest to the community working on RLHF/RLAIF and video-language alignment. However, the claims require significant revision and the evaluation needs strengthening before the findings can be considered reliable.

**Broader Impact Concerns:**

The reliance on a proprietary closed-source API (GPT-4o) as a core component does raise minor reproducibility concerns, which are noted in the Requested Changes above.

**Claims And Evidence:**

Yes

**Claims Explanation:**

The submission contains significant gaps between its claims and the supporting evidence.

- The paper claims that traditional methods like PPO are incompatible with ranked feedback. This is inaccurate, ranks can be trivially converted to pseudo-rewards, and the paper's own Section 6.1 ablation demonstrates this works nearly as well.
- The authors attribute DPO underperformance to distribution shifts without providing theoretical or empirical evidence of this shift. Given a substantial body of successful DPO alignment work in the video domain (VideoPASTA [1], RRPO [2], VistaDPO [3], LLaVA-Hound-DPO [4], TPO [5]), and the fact that the Oracle-DPO baseline appears to only use extreme best/worst responses from 5 ranked candidates rather than multiple permutation pairs or listwise approaches (e.g., LiPO [6]), this comparison is both unsupported and unfair.
- The cost-effectiveness claims are overstated: the implementation requires querying GPT-4o to rank 5 candidate responses per RL step, which at scale is considerably more expensive than a local reward model forward pass, and reliance on a closed-source API with silent version changes undermines reproducibility.
- All empirical validation is conducted on a LLaMA-2-7B-based architecture, which limits the generalizability of the broad claims made about advancing multi-modal video model fine-tuning.

[1] VideoPASTA: 7K Preference Pairs That Matter for Video-LLM Alignment; EMNLP 2025

[2] Self-alignment of Large Video Language Models with Refined Regularized Preference Optimization; NeurIPS 2025

[3] Video Hierarchical Spatial-Temporal Direct Preference Optimization for Large Video Models; ICML 2025

[4] Direct Preference Optimization of Video Large Multimodal Models from Language Model Reward; NAACL 2025

[5] Temporal Preference Optimization of Large Multimodal Models; arXiv:2501.13919

[6] LiPO: Listwise Preference Optimization through Learning-to-Rank; NAACL 2025

**Requested Changes:**

**Critical**:
1. Remove or correct the claim that PPO and traditional RL methods are incompatible with ranked feedback. Reframe GRPO_rank accurately as a specific parameterization of GRPO using nDCG reward shaping, rather than a fundamentally new algorithm.

2. Either provide concrete empirical or theoretical evidence of the distribution shifts claimed to affect DPO in this context, or remove the claim entirely. Additionally, the DPO baseline construction must be detailed: explain why only extreme best/worst pairs were used, and why multiple permutation pairs or listwise DPO approaches (e.g., LiPO) were not included as baselines.

3. Rephrase all cost-effectiveness claims to acknowledge the non-trivial financial cost of GPT-4o API queries at RL-training scale. To substantiate the claim of an efficient drop-in replacement, provide an ablation using an open-weights multimodal model as the Oracle ranker (e.g., Qwen3-VL-8B or InternVL3.5-8B), which would make the cost argument concrete and the setup reproducible.

4. The paper omits a substantial body of concurrent work demonstrating successful DPO-based video alignment including VideoPASTA (EMNLP 2025), RRPO (NeurIPS 2025), VistaDPO (ICML 2025), LLaVA-Hound-DPO (NAACL 2025), and TPO (arXiv:2501.13919) none of which are cited despite being directly relevant. These omissions weaken the paper's motivation. Additionally, LiPO is not evaluated as a baseline. Given that 5 ranked responses are available per training step, the authors should either include LiPO or a multi-pair DPO construction as a baseline, or clearly justify why Oracle-DPO was limited to only the extreme best/worst pair.

**To strengthen the work**:
The first evaluation (MSVD, MSRVTT, ActivityNet) relies on GPT-as-judge scoring on a 0–5 scale, which carries known variance and reproducibility limitations. The paper would benefit substantially from including standard contemporary benchmarks used consistently across concurrent methods: TempCompass, MVBench, MLVU, MMVU, LongVideoBench, and VideoHallucer / VidHalluc (hallucination). This would enable direct comparison with related work and allow the community to assess the framework's actual standing. Additionally, explicitly stating in the limitations that validation has only been conducted on a LLaMA-2-based architecture, with scaling behavior on stronger base models (Qwen3-VL, InternVL3.5) left as future work, would strengthen the paper's credibility.

---

> ### Author Response · Authors · 2026-04-28
> **Response to Reviewer h6e6**
>
> Changes for Reviewer h6e6 are in **blue** in the revised paper.
>
> ### **Critical Change 1: Algorithm Characterization**
>
> We would like to clarify that we did not mean that PPO is impossible to apply/completely incompatible with ranked feedback (as evidenced by we adding it as a baseline for the empirical evaluation). We actually meant that reducing a feedback that comes in the form of a ranked list into a reward function is less than ideal. Our empirical evaluation confirms the fact that our method that explicitly handle the ranked feedback outperform the "normal" PPO approach. We adjusted the terminology in the paper to make this more clear. **[See: Fine-tuning Policy Models using Oracle Ranking (paragraph 1)]**
>
> As suggested we have toned down a bit the claimed contribution by highlighting that `GRPO_rank` is a modification of the GRPO framework using nDCG-based reward shaping, rather than a completely new algorithm. **[See: Abstract, Related Works (paragraph 3)]**
>
> ### **Critical Change 2: DPO Baselines & Listwise Evaluation**
>
> We have removed the claim of distribution shifts impacting DPO's performance. Regarding the DPO baseline: we would like to clarify that our initial implementation sampled **two** responses per prompt to reflect standard pairwise DPO practices, rather than "only using extreme best/worst responses from 5 ranked candidates" (but still keeping the same total number of updates). **[See: Evaluation Protocol (Paragraph 3)]**
>
> While we agree that incorporating **LiPO** as a baseline would emphasize `GRPO_rank's` usefulness, we were unable to implement and add those extra experiments in the two weeks for the rebuttal. We have listed the comparison to this type of approach as future work. **[See: Conclusion]**
>
> ### **Critical Change 3: Oracle Costs & Reproducibility**
>
> Multiple reviewers expressed concerns about the reproducibility and cost of running closed models as oracles. To address both concerns we added to our experiments our framework using an open weight model (**PaliGemma 2 [10B]**) as the Oracle ranker. As reported in the results, the open-source model achieved overall higher performance than the SFT model but lower performance than the GPT oracle. This demonstrates that: (i) performance gains can be achieved even with lower-cost models; (ii) the more capable the oracle, the higher the expected performance gains. **[See: 6.1 Replacing the Oracle Ranker with PaliGemma2]**
>
> ### **Critical Change 4: Related Work**
>
> We thank the reviewer for the suggestion of several concurrent works in video alignment. We have updated our Related Work section to include and discuss *VideoPASTA, RRPO, VistaDPO, LLaVA-Hound-DPO,* and *TPO*. **[See: Related Works]**
>
> **Benchmarking & Generalization**
> We agree with the reviewer's concern regarding the variance of GPT-as-judge scoring on traditional datasets (MSVD, MSRVTT, ActivityNet). However, for those datasets specifically there is no better way to directly match model outputs to the groundtruth captions, as lexical matching would be very inappropriate. On the other hand, Video-MME expects multi-choice answers not depending on a model judge so it could be seen as a more reliable metric. **[See: Evaluation Datasets and Benchmarks, Conclusion]**
>
> **Architectural Limitations & Scaling**
> We have updated the `Evaluation Protocol` section to explicitly state that our current findings are based on the **LLaMA-2** architecture. We also agree that investigating the scaling behavior on more recent base models (such as Qwen3-VL or InternVL3.5) is a vital direction. While we did not have time to complete these ablations, we mention them for future work in our conclusion. **[See: Evaluation Protocol, Conclusion]**
>
> We believe these points and revisions significantly enhance the paper's credibility and its utility to the community.

---

> > ### Comment · Reviewer_h6e6 · 2026-05-08
> >
> > Thank you for the changes.
> > Because the submission now presents a clear narrative where the claims match the evidence, I am updating my recommendation.

---

### Review · Reviewer_dA6D · 2026-04-08

**Summary Of Contributions:**

The paper proposes Oracle-RLAIF, a reinforcement-learning-from-AI-feedback framework for video-language models that replaces a trained scalar reward model with an external oracle ranker that only needs to rank candidate responses, making the pipeline simpler and potentially more flexible. It also introduces GRP Orank, a rank-based variant of GRPO that converts oracle-provided rankings into rank-aware advantages for policy optimization. Empirically, the paper reports that this framework improves a supervised fine-tuned video model over prior fine-tuning baselines on several video understanding benchmarks, and includes ablations on using an API oracle versus a trained reward model and on the number of sampled candidate responses.

From my perspective, rank-based rewards may actually introduce additional randomness. For example, when multiple responses differ only in minor ways (e.g., (1) A panoramic video of the Statue of Liberty in NYC (2) A sweeping video shot of the Statue of Liberty in New York.), a rank-based reward still has to force a distinction among them, even when those differences may not be meaningful. This could be counterproductive and may add unnecessary noise to training.

**Audience:**

Yes

**Audience Explanation:**

I think this new way of designing RL objectives for RLHF/RLAIF will be of interest to the community.

**Claims And Evidence:**

No

**Claims Explanation:**

For MSVD/MSRVTT/ActivityNet, evaluation relies on GPT-4o as judge, while GPT-4o is also used as the oracle ranker during training.

The paper describes GRP Orank as a “policy optimization algorithm that directly integrates ranking feedback,” but to me it reads more like a practical modification of GRPO that uses rank-based penalties, rather than a well-motivated objective with strong theoretical justification. The math section shows some nice properties, like boundedness and a zero-sum structure, but that alone does not really show that this is the most appropriate or effective objective for optimizing ordinal feedback.

**Requested Changes:**

- Can the author check if training and evaluation using different models?
- Could GPT-4o reliably provide oracle rankings? It would be helpful if the authors could provide some evidence for this assumption, since I personally suspect that GPT-4o-generated oracle rankings may contain substantial noise, which could in turn introduce additional instability into RL training.
- Will chance the oracle ranker used in training may performance changes?

---

> ### Author Response · Authors · 2026-04-28
> **Response to Reviewer dA6D**
>
> Changes for Reviewer da6D are highlighted in **orange** in the manuscript.
>
> ### **Addressing Requested Change 1**
> > *Can the author check if training and evaluation using different models?*
>
> We would like to clarify that we had a typo in the previous version of the manuscript. The models used for Oracle in training and for evaluation were actually **DIFFERENT MODELS**, more specifically GPT-4o and GPT-3.5-Turbo. Multiple reviewers pointed out correctly that it would be inappropriate to use the same model for both, but that was not the case for our evaluation. We apologize for the confusing typo. We have corrected all mentions to the models throughout the manuscript.
>
> ### **Addressing Requested Change 2**
> > *Could GPT-4o reliably provide oracle rankings? It would be helpful if the authors could provide some evidence for this assumption, since I personally suspect that GPT-4o-generated oracle rankings may contain substantial noise, which could in turn introduce additional instability into RL training.*
>
> We appreciate the Reviewer's concern regarding the potential for noise in GPT-4o-generated rankings. While we did not have the budget to perform an in-depth empirical evaluation of GPT's ability in ranking captions, we refer to **recent literature** and the construction of our **loss function** to rebut the claim that GPT-4o would create substantial noise and instability in RL training.
>
> 1. **Recent Literature:** Lee et al. (2024) introduced Prometheus-Vision, demonstrating that multimodal judges achieve a Pearson correlation of 0.87 with human experts on video language tasks. Furthermore, in the paper *Self-Improving VLM Judges Without Human Annotations* (Lin et al. 2025), GPT-4o serves as a foundational benchmark, achieving a performance of 60.1% on the VL-RewardBench. This result validates that GPT-4o possesses a robust ability as a judge to accurately distinguish between high-quality and low-quality multimodal natural language responses.
>
>    Regarding the concern that rank-based rewards force a distinction between nearly identical responses, we reference RewardBench (Lambert et al., 2024). The study identifies that the primary issue of current trained reward models is an inability to resolve "hard negatives"—responses that appear similar but differ in subtle reasoning. Identifying the difference between a "panoramic" vs "sweeping" description is exactly the nuanced alignment that separates a base model from a high-performing one. **[See: Detailed Analysis ("We hypothesize that VLM-SFT slightly outperforms...")]**
>
> 2. **Loss Function Robustness (nDCG):** Our use of the **nDCG** metric within the $GRPO_{rank}$ loss specifically reduces the impact of "rank noise." Because nDCG applies a logarithmic discount to lower-ranked items, the optimization is primarily driven by the Oracle's ability to identify the highest-quality candidates, which is the task where GPT-4o is most stable. **[See: Mathematical Properties (Position-Sensitive Discounting)]**
>
> ### **Addressing Requested Change 3**
> > *Will changing the oracle ranker used in training may performance changes?*
>
> As further explained in the response to R1, we added an ablation with **PaliGemma 2 [10B]** as the Oracle ranker which should help answer the question on whether changing the model used as oracle would impact performance. As one would expect, the higher-quality oracle results in higher performance gains; however, even the "weaker" oracle outperforms the baseline VLM-SFT model. **[See: 6.1 Replacing the Oracle Ranker with PaliGemma2]**
>
> We believe these points and revisions significantly enhance the paper’s credibility and its utility to the community.

---

### Review · Reviewer_yyYV · 2026-04-15

**Summary Of Contributions:**

This paper proposes Oracle-RLAIF, a framework that replaces the trained reward model in video VLM fine-tuning with a general-purpose Oracle ranker (GPT-4o). Instead of learning scalar rewards, the Oracle ranks candidate responses by quality. A new loss function called $GRPO_{rank}$, a GRPO variant with nDCG-based advantage, optimizes the policy from this ordinal feedback. Experiments on MSVD, MSRVTT, ActivityNet, and Video-MME show improvements over VLM-RLAIF and other baselines.

**Strengths**: Practical idea of using a drop-in API ranker instead of training a reward model. Useful ablations (API vs trained RM, number of return sequences). Analysis of $GRPO_{rank}$'s mathematical properties. Use of Video-MME (multiple choice) as an objective evaluation benchmark.

**Weaknesses**: GPT-4o serves as both Oracle (training) and judge (evaluation), raising circular bias concerns. Oracle-DPO underperforms SFT without adequate explanation. VLM-SFT is missing as a baseline in the Video-MME evaluation (Table 2).

**Additional Comments:**

N/A

**Audience:**

Yes

**Audience Explanation:**

Integrating rank-based feedback into GRPO is timely given GRPO's central role in LLM post-training since DeepSeek-R1. The empirical finding that an API-based Oracle matches a trained RM (Figure 3) is practically useful. The Video-MME category-level analysis (Table 2) offers fine-grained insights into which video understanding capabilities benefit most from rank-based optimization.

**Broader Impact Concerns:**

No major concerns. The reliance on closed-source GPT-4o as Oracle limits reproducibility, and Oracle biases may propagate to the policy model. Experiments with open-source Oracle models would help mitigate this.

**Claims And Evidence:**

No

**Claims Explanation:**

1.  **GPT-4o's dual role creates potential circular bias.**
In Table 1, GPT-4o acts as the Oracle (ranking 5 candidate responses) during training, and simultaneously as the LLM-as-a-judge (scoring 0–5) during evaluation. Because the policy model learns GPT-4o’s specific preference patterns (e.g., vocabulary, verbosity), it naturally receives higher scores from the exact same model during evaluation. This provides an unfair structural advantage over SFT or VLM-RLAIF, which rely on their own trained RMs.


2. **Oracle-DPO underperforms SFT.**
Oracle-DPO scores below VLM-SFT on multiple benchmarks (MSVD, ActivityNet). Typically, applying DPO on top of SFT is expected to yield improvements, so this degradation is surprising. Since this baseline is used to motivate $GRP_{Orank}$'s advantage over DPO, a brief analysis of why DPO fails here would strengthen the comparison.



3. **VLM-SFT is missing from Table 2.**
Table 2 (Video-MME) compares Oracle-DPO, VLM-RLAIF, and Oracle-RLAIF, but omits VLM-SFT. Since RL fine-tuning should improve upon the SFT starting point, this baseline is essential. VLM-RLAIF scores 36.9%, and without VLM-SFT as a reference, it is unclear whether this even exceeds SFT performance. The actual gain attributable to Oracle-RLAIF cannot be properly assessed without this comparison.

**Requested Changes:**

**Major**
1. Acknowledge and discuss the GPT-4o dual-role issue (Oracle during training, judge during evaluation). Consider emphasizing the Video-MME results, which are free from this concern, as the primary evidence for the paper's claims.
2. Briefly discuss why Oracle-DPO underperforms SFT, to strengthen the motivation for $GRPO_{rank}$.
3. Include VLM-SFT as a baseline in Table 2 (Video-MME).
4. Provide a concrete cost comparison between API calls and RM training, accounting for RM reusability.
5. Specify the temperature and sampling strategy for candidate generation. Clarify what inputs the Oracle receives (images only, or also video narrative context). Also, the number of training epochs for Oracle-RLAIF does not appear to be stated.


**Minor**
- Consider including more relaetd works including recent ranking-based alignment methods: (a) listwise preference optimization (LiPO, PPA); (b) recent video VLM alignment (RRPO, NeurIPS 2025; LLaVA-Hound-DPO, NAACL 2025; RLAIF-V, CVPR 2025; iterative preference optimization methods); (c) GRPO extensions (DAPO, PREF-GRPO); (d) current RM-free alignment (DPO variants, RLVR, DeepSeek-R1-Zero).
- Justify the non-standard DCG definition (Eq. 11) and compare with simpler rank-based alternatives. Qualify the "first" novelty claim as needed.
- Eq. 5: missing definitions after "where:" (μ_R(q), σ_R(q)).
- p.2: "acheives" → "achieves".
- p.10: near-duplicate sentence ("These improvements results point to...").
- Table 2 Δ values do not match the text in several places (e.g., Temporal Perception: +21.2% in text vs +20.2% in table; Action Recognition: +11.7% in text vs +15.0% in table).

---

> ### Author Response · Authors · 2026-04-28
> **Response to Reviewer yyYV**
>
> Changes for Reviewer yyYV are highlighted in **purple** in the revised paper.
>
> ### **Addressing Major Change 1**
> > *Acknowledge and discuss the GPT-4o dual-role issue (Oracle during training, judge during evaluation). Consider emphasizing the Video-MME results, which are free from this concern, as the primary evidence for the paper's claims.*
>
> As mentioned in response to R2, we regretfully had a typo in the first version of the manuscript, where it appeared that we used the same model for training and as the evaluation judge. We actually used **DIFFERENT** models: specifically GPT-4o and GPT-3.5-turbo. Therefore, our evaluation did not suffer from the bias that would incur from using the same model for both stages. We have updated all mentions to the models in the manuscript to reflect this.
>
> ### **Addressing Major Change 2 and 3**
> > *Briefly discuss why Oracle-DPO underperforms SFT, to strengthen the motivation for $GRPO_{rank}$. Include VLM-SFT as a baseline in Table 2 (Video-MME).*
>
> We added **VLM-SFT** as a baseline for Table 2. **[See: Detailed Analysis (Paragraph 2)]**
>
> Regarding why Oracle-DPO underperforms SFT: we hypothesize that DPO does not natively incorporate a ranking into the cost function and does not utilize the full range of candidate responses as $GRPO_{rank}$ does. Decomposing the feedback into 1-to-1 comparisons provides less information per update. While we expect DPO would eventually surpass SFT given additional optimization steps, it lacks the sample efficiency of $GRPO_{rank}$ within the same training budget. **[See: Detailed Analysis (Paragraph 1)]**
>
> ### **Addressing Major Change 4**
> > *Provide a concrete cost comparison between API calls and RM training, accounting for RM reusability.*
>
> Estimating exact costs for Reward Model (RM) training is challenging as the original VLM-RLAIF team did not report specific training times or the details of creating the video narratives required. However, using the data available, we provide the following estimates:
>
> * **Oracle-RLAIF (GPT-4o):** Incurs a cost of approximately **$128 per epoch** for our 99k dataset.
> * **Local RM (13B):** We estimate the one-time cost of creating video narratives and training the RM at **$250** (based on current infrastructure/API pricing).
>
> Price parity with API calls is achieved after roughly **two epochs**. While the RM becomes "cheaper" over time, it requires additional GPU infrastructure to maintain and use. To reinforce that Oracle-RLAIF remains cost-efficient and flexible, we performed an additional experiment using **PaliGemma 2 [10B]** as an open-source Oracle. This demonstrates that our framework remains effective even for users without the budget for high-end API calls. **[See: 6.1 Replacing the Oracle Ranker with PaliGemma2]**
>
> ### **Addressing Major Change 5**
> > *Specify the temperature and sampling strategy for candidate generation. Clarify what inputs the Oracle receives (images only, or also video narrative context). Also, the number of training epochs for Oracle-RLAIF does not appear to be stated.*
>
> We have added all requested hyperparameters to the manuscript:
> * **Temperature:** 0.8
> * **Strategy:** We use `num_return_sequences=5` via the HuggingFace Transformers library, allowing the model to explore 5 different paths.
> * **Training Epochs:** 2 (over a dataset of 99k videos).
>
> **Oracle Inputs:** To rank candidate responses, the Oracle receives:
> 1. The same images used by the VLM for inference.
> 2. The five candidate responses.
> 3. The specific prompt/instructions for ranking.
>
> Notably, the **Oracle ranks without video narrative context**, relying on the visual frames provided. **[See: Evaluation Protocol]**
>
> ### **Addressing Minor Changes**
> We have added **LiPO** (Zheng et al., 2025), **RRPO** (Sarkar & Etemad, 2025), **VideoPasta** (Kulkarni & Fazli, 2025), **TPO** (Li et al., 2025), and **VistaDPO** (Huang et al., 2025) to our Related Works or Conclusion sections. All other minor editorial changes have been addressed.
>
> ---
>
> We believe these points and revisions significantly enhance the paper’s credibility and its utility to the community.

---

> > ### Comment · Reviewer_yyYV · 2026-05-11
> >
> > Thank you for the thorough revision. My main concerns have been resolved, and I am updating my recommendation.

---

### Decision · Action_Editor_3kiD · 2026-05-31

**Recommendation:** Accept with minor revision

**Additional Comments:**

For the minor revision, the authors should make sure that the final manuscript clearly reflects the more modest and accurate framing established during the rebuttal. In particular, GRPO-rank should be described as a rank-aware modification of GRPO rather than as a fundamentally new policy optimization algorithm. The limitations should explicitly state that oracle reliability is assumed rather than comprehensively validated, that closed-source oracle rankers introduce reproducibility concerns, and that the main experiments are limited to a LLaMA-2-based VLM architecture. The paper should also clearly distinguish evidence from GPT-judge-based benchmarks and evidence from more objective multiple-choice evaluation such as Video-MME. Finally, the authors should include a clear discussion of missing listwise and multi-pair preference baselines, such as LiPO or stronger DPO variants, as future work rather than implying that the current Oracle-DPO comparison exhausts the relevant baseline space.

**Audience:**

Yes

**Audience Explanation:**

The paper is likely to be of interest to at least some members of the TMLR audience. The topic connects to several active areas, including RLHF/RLAIF, preference optimization, multimodal model alignment, video-language understanding, and the use of rank-based feedback for policy optimization. The idea of replacing a trained multimodal reward model with a general-purpose oracle ranker is practically motivated, and the empirical comparison between API-based and open-weight oracle rankers is useful for researchers considering cost, flexibility, and reproducibility tradeoffs. While the methodological novelty is incremental relative to GRPO and existing preference-optimization methods, the paper offers a clear and useful study of a relevant design point in multimodal alignment.

**Claims And Evidence:**

Yes

**Claims Explanation:**

Summary: The paper proposes Oracle-RLAIF, a framework for fine-tuning video-language models using reinforcement learning from ranked AI feedback. Instead of training a dedicated scalar reward model, the method uses an external oracle ranker to rank multiple candidate responses and then optimizes the policy using a rank-aware variant of GRPO, called GRPO-rank, based on an nDCG-inspired advantage construction. The paper evaluates the approach on several video understanding benchmarks, including MSVD, MSRVTT, ActivityNet, and Video-MME, and compares it against SFT, DPO, and RLAIF-style baselines. The main contribution is a practical rank-based RLAIF pipeline that reduces reliance on trained reward models and provides empirical evidence that listwise ranked feedback can improve video-language model alignment.

Comment: Overall, I find that the main claims of the paper are sufficiently supported for acceptance, subject to minor revision. The initial reviews raised several important concerns, including possible circularity from using the same model for training-time oracle ranking and evaluation-time judging, insufficient explanation of Oracle-DPO underperformance, missing SFT baselines in the Video-MME comparison, unclear cost and implementation details, overstatement of the novelty of GRPO-rank, and incomplete positioning with respect to recent video preference-optimization methods. The authors’ revision addressed many of these issues: they clarified that different models were used for training and evaluation, added the missing VLM-SFT baseline, provided additional implementation details and cost discussion, toned down claims about the algorithmic contribution, added an open-weight PaliGemma 2 oracle ablation, and expanded the related work. These revisions substantially improve the alignment between the claims and the evidence. Some limitations remain, particularly the dependence on oracle rankings whose reliability is not directly evaluated in depth, the limited validation on a LLaMA-2-based architecture, and the absence of stronger listwise or multi-pair preference baselines. However, these limitations are now sufficiently acknowledged and do not undermine the core empirical claim that rank-based oracle feedback can be a useful practical alternative to trained reward models in this setting.